# Sneaking into a Hotspot of Biodiversity: Coverage and Integrity of a Rhodolith Bed in the Strait of Sicily (Central Mediterranean Sea)

Teresa Maggio [1,*], Patrizia Perzia [1], Alfredo Pazzini [2], Silvana Campagnuolo [3], Manuela Falautano [1], Anna Maria Mannino [4] and Luca Castriota [1]

1. Department of Monitoring, Environmental Protection and Biodiversity Conservation, Italian Institute for Environmental Protection and Research (ISPRA), Lungomare Cristoforo Colombo n. 4521, 90149 Palermo, Italy
2. Department of Monitoring, Environmental Protection and Biodiversity Conservation, Italian Institute for Environmental Protection and Research (ISPRA), Via Vitaliano Brancati n. 48, 00144 Roma, Italy
3. Department for Environmental Assessment, Control and Sustainability, Italian Institute for Environmental Protection and Research (ISPRA), Lungomare Cristoforo Colombo n. 4521, 90149 Palermo, Italy
4. Department of Biological, Chemical and Pharmaceutical Sciences and Technologies (STEBICEF), University of Palermo, Via Archirafi n. 28, 90123 Palermo, Italy
5. GRAM Gruppo di Ricerca Applicata al Mare Soc. Coop., Via Roma 471, 90139 Palermo, Italy
* Correspondence: teresa.maggio@isprambiente.it

**Abstract:** Habitat mapping, physical characteristics and benthic community of a rhodolith bed in the Pelagie Islands (Strait of Sicily, Mediterranean Sea) were studied through Multi–Beam Echo–Sounder (MBES), Remotely Operated Vehicle (ROV) and grab samples. The geomorphological analysis revealed an articulated and wide rhodolith bed; video inspections highlighted a bed with high coverage, few sandy patches and with a prevalence of the boxwork morphotype. A total of 207 taxa with 876 specimens were identified, and Polychaeta was the dominant taxon. *Linguimaera caesaris*, a Lessepsian benthic amphipod, was recorded in all sampling sites, and its presence represents an input to deepen the benthic assemblage research on the rhodolith bed. In terms of morphotype composition, dead/live ratio and species variability, the bed variability indicated a good status of health, although trawling signs were detected through ROV videos. The present study broadens the knowledge on Mediterranean rhodolith beds and supports the importance of survey and monitoring activities for the conservation and management of this important habitat.

**Keywords:** biodiversity; community composition; rhodolith morphotypes; habitat mapping; Lampedusa Island; macroalgae; bioconstructions; *Linguimaera caesaris*

## 1. Introduction

Rhodolith beds, which also include the maërl beds, are unique bioconstructions characterized by unattached non-geniculate calcareous red algae with a worldwide distribution. Rhodolith-forming algae grow as unattached nodules, formed by at most 50% of calcareous Rhodophyceae [1], living on the sediment in the photic zone [2]. These beds occur in tropical, temperate and polar environments [3]. In Europe they are known along most of the Eastern Atlantic coast, from Portugal to Norway and throughout the Mediterranean. The depth ranges from 5 to 35 m in the Western Atlantic and at greater depths (up to 150 m) in the Mediterranean Sea [4–7]. Mediterranean rhodolith/maërl beds, considered as "facies" of infralittoral and circalittoral biocoenosis [8–10], were reported for almost all sectors except for the Eastern Adriatic, Libyan, Egyptian, Syrian and Lebanon coasts; they are not even reported in the Black Sea [4]. Rhodolith/maërl beds can be typically found around islands and seamounts, marine terraces, channels and banks, mainly in the mesophotic

zone. Knowledge on their distribution is not homogeneous, but is more concentrated in some areas where specific research or monitoring activities are carried out.

In the Mediterranean Sea, rhodolith/maërl beds develop on coarse sand and fine gravel under the influence of bottom currents in the infralittoral zone, and on the coastal detritic bottom in the circalittoral zone. The algal growth in successive layers is facilitated by continuous rotation of the biogenic concretions on a substrate influenced by currents [11,12]. Rhodolith/maërl beds show a spatial complexity with a high level of floral and faunal biodiversity [7]; algae forming rhodoliths have been described as ecological engineers since they create many ecological niches for different species, allowing a great deal of interstitial life and favoring the settlement of a variety of sessile organisms and endobionts. As they are considered both hard and soft substrata, rhodolith/maërl beds are among the most biodiverse communities of the Mediterranean Sea after *Posidonia oceanica* meadows and coralligenous habitat; they encounter about 1000 different species belonging to macrobenthos, 70% of animals and 30% of plants [13]. Furthermore, rhodolith/maërl beds provide nursery grounds for commercial species of fish and shellfish, maintaining sustainable fisheries [14,15] and ensuring a wide spectrum of ecosystem goods and services [14,16].

Rhodolith/maërl beds are considered to be a non-renewable resource because of the slow growth rate (1 mm/year), coupled with the high rate of destruction and extraction by anthropic activities [12,17]. They are particularly sensitive habitats to several pressures and impacts such as maritime traffic, fishery, and sea bottom alteration such as dredging, abrasion/mechanical damage, water pollution, temperature raising and invasive alien species [8,18]. In particular, negative effects of fishing trawl activities have been widely demonstrated: rhodolith/maërl bed are broken and biodiversity decreases [15,19–21].

These pressures and impacts, coupled with long periods of habitat recovery, render the rhodolith/maërl bed very vulnerable so that it is necessary to implement management and protection measures to avoid its degradation. Some authors studied the effects of different fishery management measures to preserve the rhodolith/maërl bed status; in particular, Consoli et al. [21] indicated fishery spatial restrictions, such as no-take zones, as the more effective management tool to protect it rather than technical measures (e.g., increase of trawl net mesh size). Farriol et al. [22] reported signals of recovery of rhodolith/maërl beds in areas closed to fishery, and demonstrated the effectiveness of no-take zone measures for the conservation of the habitat. The main conservation and management measures are found in the Barcelona Convention's Action Plan, where coralligenous and rhodolith/maërl assemblages, together with *P. oceanica* meadows, are indicated to require a granted legal protection, although it is not legally binding [23]. Moreover, within the European legislation, the need for protection of this particular habitat has been pointed out in the Habitats Directive (1992/43/EC) [24], the EC Council Regulation 1967/2006 [25] and the Marine Strategy Framework Directive 2008/56/EC (MSFD) [26]. In particular, the Habitat Directive, whose main goal is the conservation of biodiversity through natural habitat and (rare, threatened or endemic) animal and plant species protection, included the two key species forming the maërl bed, *Lithothamnion corallioides* (P. Crouan & H. Crouan) P. Crouan & H. Crouan, 1867 and *Phymatolithon calcareum* (Pallas) W.H. Adey & D.L. McKibbin ex Woelkering & L.M. Irvine, 1986 in the Annex V. However, unfortunately the habitat as such is not included. The EC 1967/2006, concerning fishery measures for a sustainable use of the resources in the Mediterranean Sea, banned trawl fishery on the maërl bed. Lastly, given the ecological importance of the rhodolith/maërl bed and the lack of knowledge on its presence and status along the coasts, within the Descriptor 1 "Biodiversity" of MSFD, it was proposed to monitor the rhodolith/maërl bed extent as proxy of the presence and health status of the protected species *L. corallioides* and *P. calcareum*.

Rhodolith/maërl bed monitoring represents the basic tool to implement management and conservation measures. Unfortunately, the scarcity of relevant geospatial and ecological data, slowed the effective application of conservation and management measures. In the last period, rhodolith/maërl bed knowledge was deepened thanks to the extensive regional-scale monitoring program of MSFD, as well as several research activities carried

out on it [27–29]. In this context, within the objectives of the Interreg Italia-Malta, named HARMONY, rhodolith beds in the Strait of Sicily were investigated. In particular, the rhodolith/maërl bed spatial extent, physical characteristics, and associated macrobenthos diversity were studied in the Pelagie Islands, according to the monitoring plans developed in the MSFD.

## 2. Materials and Methods

### 2.1. Study Area

The study area is located in the southeast off Lampedusa Island (Figure 1). It is the largest island of the Pelagian archipelago, located in the Strait of Sicily (Mediterranean Sea). It is entirely made of sedimentary rocks and shows two main sectors: the northern sector, with dominant coastal features varying from steep to sub-vertical cliffs, and the southern sector with the coast gradually sloping down and including several small coves. The south eastern area off Lampedusa is characterized by a coarse to fine sand bottom with *P. oceanica* meadows as the main habitat close to the coast, followed by sandy seabeds at increasing depth [30].

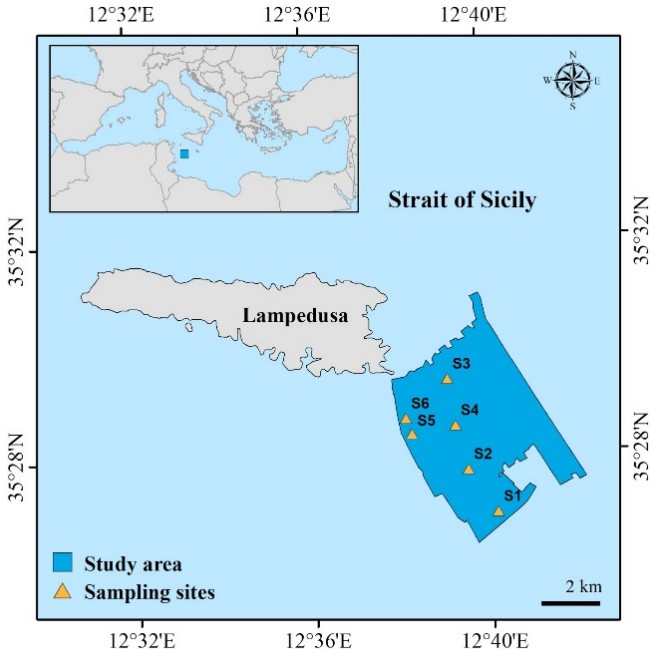

**Figure 1.** Map of Lampedusa study area with indication of sampling sites (S1–S6).

A survey was carried out in May 2019 on board the Research Vessel Astrea of the Italian Institute for Environmental Protection and Research (ISPRA). This was part of the activities of the HARMONY project, collecting data on seafloor integrity, rhodolith/maërl bed status and associated macrobenthic community. The investigations were carried out at depths from 35 to 100 m, based on literature information [21] as well as on indications provided by fishermen on the presence of the rhodolith/maërl bed. A sedimentary bottom characterized by unattached non-geniculate calcareous red algae was considered a "rhodolith bed" when the coverage was >10% of living rhodoliths and its extension was at least 500 m$^2$ [8].

### 2.2. Habitat Characterization: Analysis of Acoustic and Video Data

Acoustic data were used to define morpho-bathymetry of the investigated area (Figure 1) through the Kongsberg EM2040 multibeam system managed by the SIS (Seafloor Information System) software. Caris HIPS & SIPS 9.1 software (Teledyne, Geospatial, Fredericton, NB, Canada) was used to process multibeam data, backscatter and the related mosaic. Multibeam geophysical data were acquired at a frequency of 300 kHz using an

average swath angle of 60°, and data acquisition was carried out at an average navigation speed of about 6 knots. The acquisition strategy involved the execution of 21 parallel transects, arranged in the NW–SE direction (on average 6 km long) and 6 transects arranged in the SW–NE direction (2 km long) (approximately 140 linear km of relief). Multibeam and backscatter data were integrated into ArcGIS v.10.3 software, and were processed in order to derive the spatial distribution of terrain variables. The shaded relief (hillshade), the contours (isolines), the angle of slope and the steepest downslope direction (aspect) were elaborated using the Surface tools into the Spatial Analyst toolbox, and the roughness using the BTM (Benthic Terrain Modeler). Maps of terrain variables (bathymetry, slope, aspect and roughness) were obtained with a 1 m grid resolution, for the sediment characterization map from backscatter data. The distribution of backscatter intensity allowed to distinguish the granulometric nature of the superficial sediment with a good approximation: high values of backscatter indicate rocky areas or areas with coarse sedimentation (gravelly–pebbly), while low values are associated to the presence of medium–fine superficial sediments (silty–pelitic).

During the surveys, video images and sea-bottom samples were collected as ground truth information. Video transects were acquired through a Remotely Operated Vehicle (ROV FOII) equipped with a high-definition camera 1920 × 1080, illuminator of 13,000 lumen, two parallel laser beams at the fixed size of 10 cm for the automatic image scale estimation. According to the monitoring protocol of MSFD [31], for each sampling site (S1–S6) three 200 m long routes (transects) were carried out, at least 100 m distance from each other. ROV videos were displayed with QuickTime Player vers7 software for macOS, and a visual assessment of the bottom status was carried out; for each route, 20 video frames were extrapolated at regular time intervals in order to characterize the rhodolith/maërl bottom.

An assessment of the rhodolith/maërl bed status was conducted from videos and frames, evaluating the total percentage of cover, the main morphotype coverage (pralines, boxwork and unattached branches), and the ratio between dead and live thalli [8,32]. A percent contribution of each morphotype coverage from the videoframes was plotted on a ternary diagram to show the morphotype variability among the sampling sites. The vitality of thalli was determined on the base of the observation of rhodolith coloration (pink, purple, or reddish color) following Bahia et al. [33]. In the presence of a mixed composition of rhodoliths with bare sand and/or lamellar thalli, their percent of coverage was reported; furthermore, among the analyzed videoframes, white fragments of calcareous algae were also encountered in the percent of coverage, because of its importance as substrate for the rhodolith/maërl bed.

PERMANOVAs were run on the Bray–Curtis similarity matrix of videoframes calculated on square-root transformed data, to investigate the variation in morphotype composition and in live/dead ratio between sampling sites. The analyses were run using 999 permutations of residuals under a reduced model. In case of significant differences found, PERMANOVAs were repeated to investigate pairwise comparisons between sampling sites. Changes in morphotype composition and live/dead ratio were visualized using principal coordinates (PCO) analysis on Bray–Curtis similarity matrices. A correlation vector based on Pearson ranking (>0.6) was overlaid on the PCO to visualize the relationship among the sites and ordination axes. Statistical analyses were carried out using PRIMER v6 (Primer–E Ltd., Plymouth) with PERMANOVA add-on software [34,35].

### 2.3. Biological Communities Associated with Rhodolith/Maërl Bed

According to the monitoring protocol of MSFD [31], for each sampling site, three 25 L Van Veen grab samples were collected to assess rhodolith/maërl bed vitality and to study the benthic communities; calcareous algae were placed on a flat surface to identify the different morphotypes (pralines, boxwork rhodoliths, unattached branches) on the basis of main morphology, and a percentage of coverage for each of them was estimated. Collected rhodoliths were cleaned to remove sediment and epiphytic organisms, air dried for at least

48 h, wrapped in aluminum foil and stored in dark in zipper bags. Epiphytic algae were stored at $-20$ °C for subsequent analysis.

Sediment samples were used to study the benthic assemblage; they were sieved through 1 mm sieves and subsequently sorted for the identification of the macrozoobenthos. All macrofaunal and macrofloral specimens were identified to the lowest possible taxonomic level and, for macrofauna, estimates of abundance were expressed for each site as the number of individuals per sediment volume examined. The following diversity indices were computed for each sampling site: species richness (S), number of individuals (N), Simpson's index (D), Evenness (J'), Shannon's index (H') and Margalef index (d). PERMANOVAs were performed to investigate the variation in the benthic assemblage between sampling sites. A first analysis was carried out on square-root transformed macrofauna abundance data, based on a Bray–Curtis sample similarity matrix. The analysis was run using 999 permutations of residuals under a reduced model. PERMANOVA was repeated on a macrofauna and macroflora presence/absence dataset, on the basis of the Jaccard measure. In case of significant differences, PERMANOVAs were repeated to investigate pairwise comparisons between sampling sites. For both datasets (abundance and presence/absence), Principal Coordinates analysis (PCO) was calculated on the basis of the Bray–Curtis similarity matrices among all pairs, and a correlation vector based on the Pearson ranking (>0.6) was overlaid on the PCO to visualize the relationship among the sites and ordination axes. Lastly, the functional diversity of the rhodolith/maërl bed was assessed by grouping species in the following feeding categories: suspension feeders, deposit feeders, deposit/suspension feeders, grazers, predators, omnivorous and others (comprising commensals, parasites); species with insufficient information were assigned to the 'unknown' category. Multivariate analysis was conducted to test for differences in the distribution of the feeding category, as total abundance and as number of species between sampling sites. All statistical analyses were carried out using PRIMER v6 (Primer–E Ltd., Plymouth, UK) with PERMANOVA add-on software [34,35].

## 3. Results

### 3.1. Terrain and Textural Analysis of Rhodolith/Maërl Bed

The multibeam survey extended for 26.5 km$^2$, highlighted a poorly articulated seabed that can be divided into three large areas: a rocky one extending to 50 m depth, a flat one degrading slowly towards the south–southeast up to 75 m depth, and a third area in the north east portion of the relief, where the seabed reaches 95 m depth and more rapidly approaches the limit of the Tunisian continental shelf.

The rocky area was characterized by a large elongated morphological rise extending southeast for about 1 km, covering an area of about 1.5 km$^2$. This morphological rise creates a bathymetric height difference of about 20 m towards the northeast (Figure 2a).

Proceeding to the northeast, another morphological rocky outcrop is present at 90 m depth. From the partial multibeam investigations it is evident that it extends eastwards for 650 m, causing a bathymetric height difference of about 15 m in the north-northeast direction (Figure 2a). The areas characterized by the presence of steep morphologies of rocky origin are highlighted in the slope and aspect maps in Figure 2b,c.

The remaining part of the seabed has a flat morphology, as evidenced in the roughness map (Figure 2d); only small corrugations of the sediment are present, derived from the complex geology of the Tunisian passive margin below.

Figure 3a shows the 1 m–resolution acoustic backscatter mosaic (sonogram) produced by the multibeam system. The backscatter analysis shows that most of the seabed responds to acoustic diffraction with average values between $-12$ and $-14$ dB. A Van Veen bucket sampling was performed in different areas corresponding to this acoustic interval, which was found to be characteristic of the rhodolith/maërl bed.

Backscatter identifies the rocky area described above and other areas with low backscatter value corresponding to fine, silty–pelitic particle size, with little or no presence of rhodoliths; a megaripples/dunes area is evident on the western edge, i.e., sedimentary

structures indicating the presence of powerful bottom currents and justifying the great abundance and the large size of the rhodoliths making up the bed.

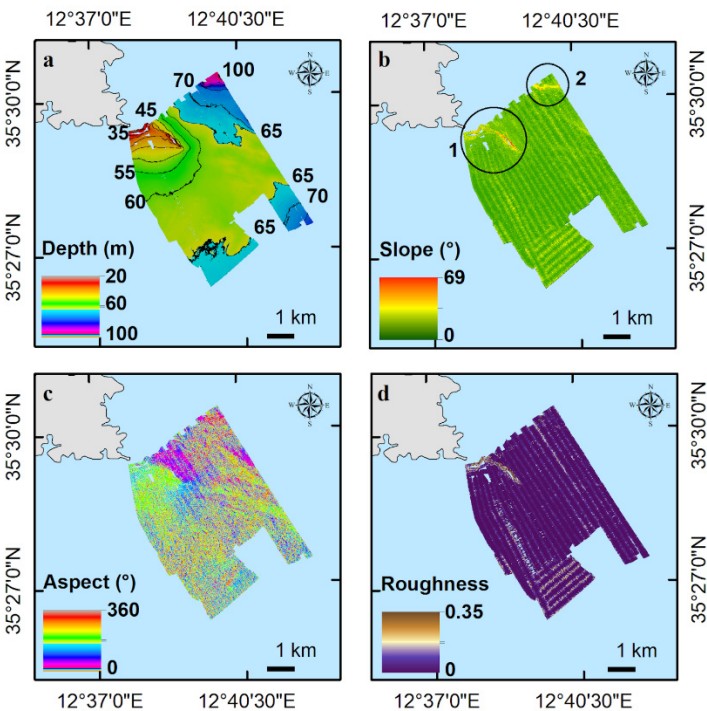

**Figure 2.** Terrain variables maps from Multibeam morpho-bathymetric surveys: (**a**) bathymetry map; (**b**) slope distribution map, in evidence the areas characterized by the presence of high morphologies; (**c**) aspect distribution map; (**d**) roughness distribution map.

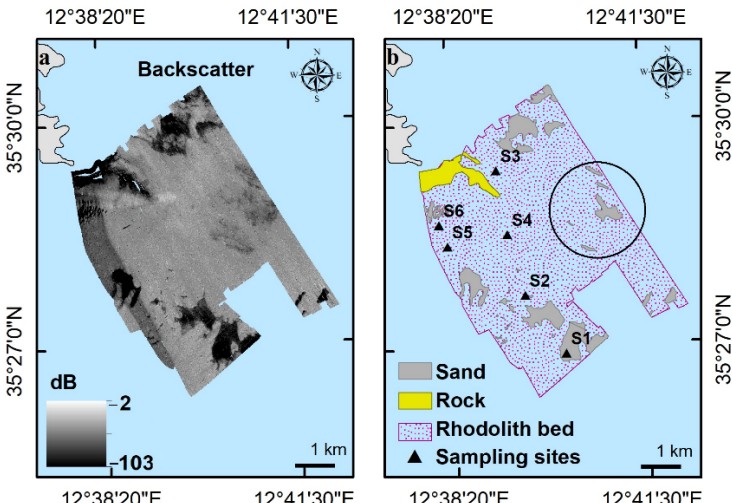

**Figure 3.** (**a**) Distribution of the backscatter intensity (sonogram) produced by the multibeam system; (**b**) simplified classification of sediments in the study area. The circle indicates an area with fewer rhodoliths; sampling sites are also reported.

The backscatter data show a silty–pelitic particle size distribution (low backscatter) on the ridges of megaripples and high values in the concave portions. These structures seem to continue along the entire eastern edge of the relief, but with less intensity and dimension. Figure 3b shows a preliminary classification of the seabed: almost 80% of the area is characterized by the presence of rhodoliths, except for the circle sector with a lower presence of rhodoliths.

### 3.2. Analysis of Video Data

Rhodoliths were found throughout the area investigated by ROV, even if with different percentages of coverage in the six sampling sites (S1–S6). Mean rhodolith cover ranged from 87% to 99%, including dead and live thalli, with the exception of S1 where the lowest values of coverage (6.2%) has been reported (Figure 4a,b). The bed was characterized by the presence of lamellar and encrusting (fouling) calcareous algae such as *Peyssonnelia* spp. (Figure 5a), with a percentage of cover ranging from 1 to 25% in S2, S3, S5 and S6. Coverage of white thalli ranged from 1% to 24% (Figures 4b and 5b), whereas the percentage of live thalli ranged from 18% to 98% with a mean value of 74%. The percentage of the three different morphotypes in the six sampling sites was highly variable among transects. It has been represented as a box plot in Figure 6a and in the ternary plot of Figure 6b. The most dominant morphotype was boxwork, except for S1 where pralines dominated; S3 showed the greatest morphotype variability.

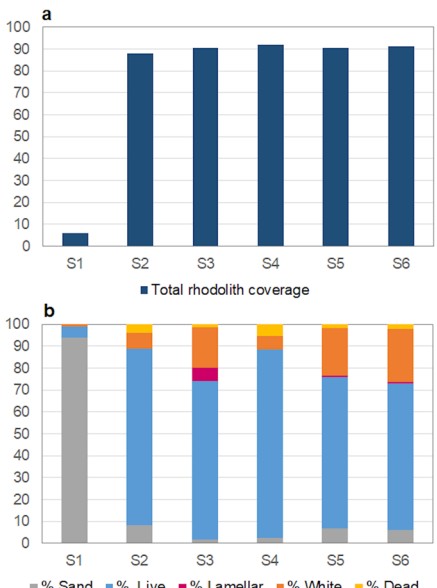

**Figure 4.** (**a**) Percentage of total rhodolith coverage in the six sampling sites; (**b**) percentage of the different components of videoframe coverage in the six sampling sites.

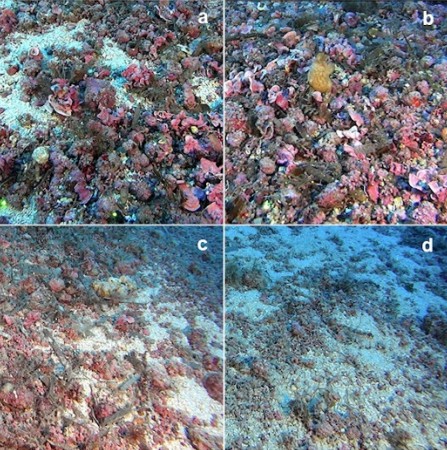

**Figure 5.** ROV image of lamellar thalli (**a,b**) in S3 sampling site and white ones in S2 (**c**) and S5 (**d**) sampling sites, among the rhodolith/maërl bed in Lampedusa Island.

The PERMANOVA performed on the morphotype dataset showed significant differences between sampling sites (Pseudo–F = 43.994; $p < 0.01$). Pairwise comparisons were

all significantly different, except for S2–S4 and S5–S6. The Principal Coordinates analysis (PCO) explained 81.8% of the variation in the data, with PCO1 axis explaining 50.6% and PCO2 axis 31.2%. S1 is separated along the PCO1 axis, while transects are separated along the PCO2 axis (Figure 7a). PERMANOVA performed on the dead/live dataset, including bare coverage, showed significant variability between sampling sites (Pseudo–F = 152.26; $p < 0.01$). Pairwise comparisons were all significantly different except for S3–S4, S3–S5 and S5–S6. Differences in dead/live ratio are visualized on the PCO graph; the 85.5% of the total variation was explained by the PCO1 axis, along which S1 separates. "Bare" correlation vector explains this diversity (Figure 7b).

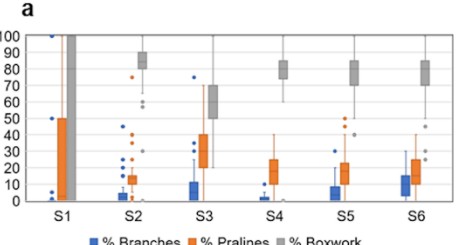

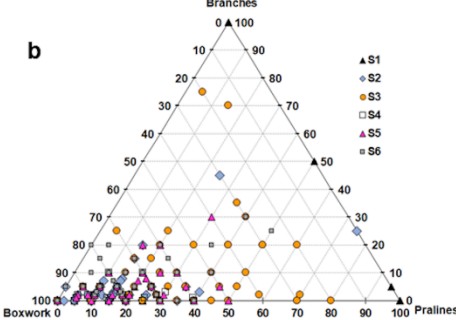

**Figure 6.** (**a**) Boxplot of three different morphotypes percentage in the sampling sites; (**b**) ternary plot of the three morphotypes (coverage in %) per sampling site.

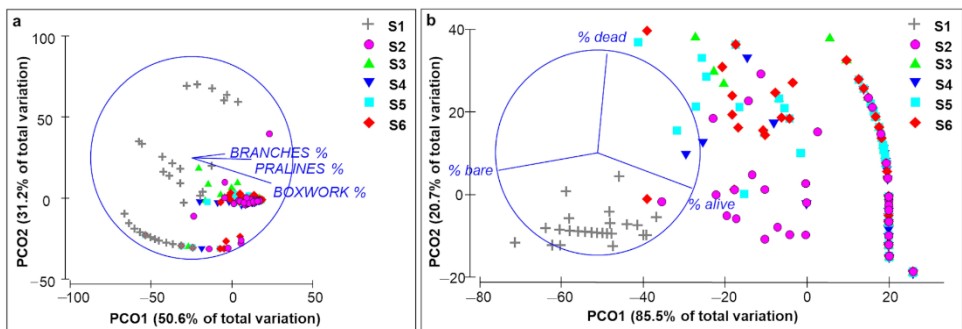

**Figure 7.** Principal Coordinates Analysis (PCO) ordination plot of the sampling sites related to morphotypes coverage (**a**) and dead/live ratio (**b**). Correlation vector is based on Pearson ranking.

### 3.3. Benthic Community Structure

Boxwork was the most represented morphotype in the grab samples, except for the S1 and S6 sampling sites, as is evident in the ternary plot (Figure 8); visual inspection detected a cover of live rhodoliths ranging from 85% to 100%.

The classification of the macrobenthos from grab samples allowed to identify a total of 206 taxa, the most diverse of which were Polychaeta (74 taxa) and Crustacea (60 taxa), followed by Algae (29 taxa) belonging to Ochrophyta and Rhodophyta, Mollusca (27 taxa), Pycnogonida (8 taxa), Echinodermata (6 taxa), Sipuncula (1 taxon) and Chordata (1 taxon) (for the list of the species, see Table S1).

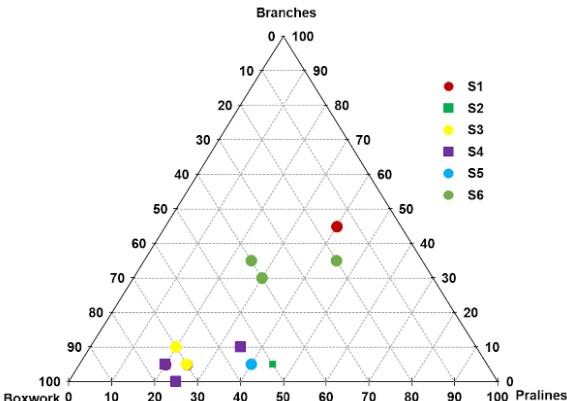

**Figure 8.** Ternary plot of the three identified morphotypes from the grab samples.

In terms of abundance, a total of 875 specimens belonging to 177 animal taxa were collected. The most abundant taxonomic groups were Polychaeta and Crustacea, with 342 and 345 specimens, respectively. The rarest taxonomic groups were Sipuncula and Chordata, with only one species each (*Phascolosoma granulatum* and *Branchiostoma lanceolatum*, respectively) and represented by eight and three individuals, respectively. The most abundant taxa were two Crustacea, the amphipod *Linguimaera caesaris* with 56 individuals (~6% of all individuals) and the isopod *Joeropsis* sp. with 38 individuals (~4% of the total). The lowest number of species was found at S1, and the lowest macrofauna abundance was recorded at S1 and S2 (Table 1).

**Table 1.** Traditional biodiversity indices of the six sampling sites. S: number of species (including macrophytes); N: number of specimens; D: Simpson index; J′: Pielou index; H′: Shannon index; d: Margalef index. Indices marked with * are calculated for macrofauna only.

| Sampling Site | S | N * | D * | J′ * | H′ * | D * |
|---|---|---|---|---|---|---|
| S1 | 29 | 42 | 0.96 | 0.93 | 3.10 | 7.22 |
| S2 | 53 | 89 | 0.98 | 0.94 | 3.66 | 10.69 |
| S3 | 100 | 190 | 0.98 | 0.93 | 4.13 | 15.82 |
| S4 | 83 | 170 | 0.98 | 0.93 | 3.97 | 13.44 |
| S5 | 99 | 198 | 0.97 | 0.90 | 3.95 | 14.75 |
| S6 | 90 | 186 | 0.98 | 0.91 | 3.90 | 13.40 |

PERMANOVA showed a significant variability on the macrofauna species abundance between sampling sites (Pseudo–F = 1.4288 $p < 0.01$). All pairwise comparisons were not significantly different. Multivariate analysis based on the macrofauna and macroflora dataset revealed significant differences between sampling sites (Pseudo–F = 1.6239 $p < 0.01$). All pairwise comparisons were not significantly different. PCO explained the 25.1% of the variation among sampling sites considering species-abundance data, and the 28.5% considering macroflora and macrofauna data (Figure 9).

Regarding the trophic group analysis, the most represented group in terms of species and individuals was that of deposit feeders followed by predators; deposit/suspension feeders and grazers were the less-represented trophic groups (Table 2). The crustacean isopod *Joeropsis* sp. was the most abundant deposit feeder, whereas the polychaetes *Glycera alba* and *Schistomeringos rudolphi* were the most abundant predators. PERMANOVA among trophic groups, both in terms of abundance and number of species, showed significant variation between sampling sites (Pseudo–$F_{abundance}$ = 2.57 $p < 0.05$; Pseudo–$F_{n\ species}$ = 2.99 $p < 0.05$).

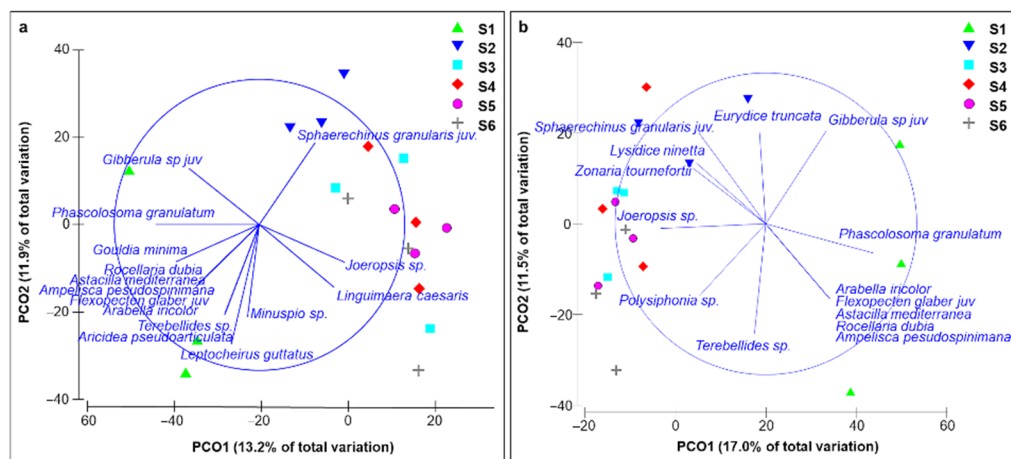

**Figure 9.** Principal Coordinates Analysis (PCO) ordination plot of the sampling sites related to macrofauna abundances among sampling sites (**a**) and related to macrofauna and macroflora presence among sampling sites (**b**). Correlation vector is based on Pearson ranking.

**Table 2.** Number of species and individuals for each trophic group.

| Trophic Group | N Species | N Species % | N Individuals | N Individuals % |
|---|---|---|---|---|
| Grazer | 9 | 5.3 | 48 | 5.8 |
| Suspension feeder | 34 | 20.0 | 118 | 14.2 |
| Deposit feeder | 51 | 30.0 | 262 | 31.6 |
| Deposit/suspension feeder | 5 | 2.9 | 78 | 9.4 |
| Omnivorous | 12 | 7.1 | 88 | 10.6 |
| Predator | 43 | 25.3 | 188 | 22.6 |
| unknown | 15 | 8.8 | 47 | 5.7 |
| other | 1 | 0.6 | 1 | 0.1 |

## 4. Discussion

The geophysical–geomorphological analysis allowed to map the sea bottom of the study area, revealing an articulated and wide rhodolith/maërl bed in the southeast area of Lampedusa Island, an area not previously investigated. The ROV survey confirmed the presence of the rhodolith/maërl bed and contributed to the habitat characterization, together with the analysis of benthic communities. Information provided by the three complementary techniques allowed to broaden the knowledge on the distribution and morphotypes composition of Mediterranean rhodolith/maërl beds, as well as providing an inventory of the associated benthic species in the Strait of Sicily. This area is a mosaic of different habitats and is considered an important biodiversity hot spot and crossroad in the Mediterranean Sea [36], being a spawning and nursery area for many commercial species [37], as well as a foraging area for predators such as the loggerhead sea turtle [38].

The rhodolith/maërl bed offshore Lampedusa Island is characterized by a rhodolith cover (>87%) higher than other Mediterranean rhodolith/maërl beds, where maximum values vary between ~55–66% in the Tyrrhenian Sea [25,39], ~43% in the Adriatic Sea, and ~66% in the Ionian Sea [40]. Furthermore, it is characterized by high percent coverage of live rhodoliths (not less than 70%) (Figure 10a), and rare sandy patches with low percent coverage of live rhodoliths (Figure 10b), as confirmed by backscatter results (Figure 3b) and videoframes analysis. In some cases, uncovered portions with white fragments of calcareous algae have been recorded; this substrate constitutes the organogenous base of the living rhodolith/maërl bed. This overall scenario highlighted the good health status of the bed, although furrows probably left by trawling were at times detected, suggesting that the area is subjected to anthropic pressures (Figure 10c).

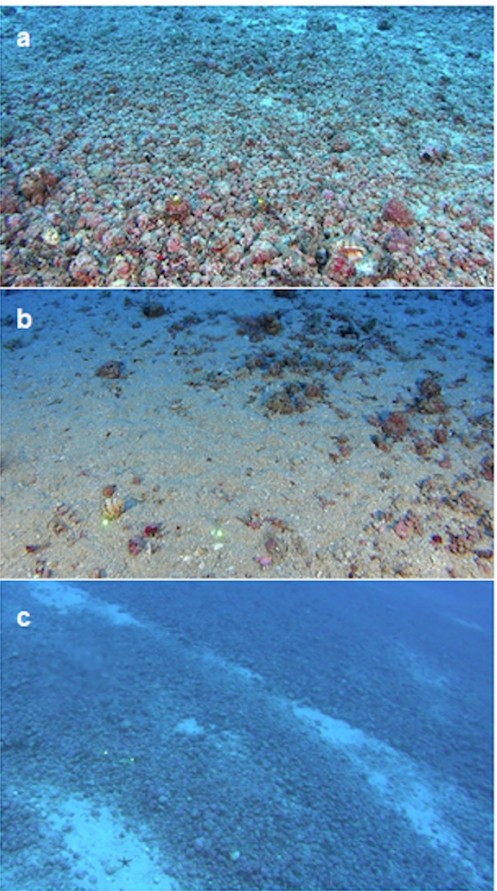

**Figure 10.** (**a**) Area characterized by high percent coverage of live rhodolith in S4; (**b**) area of rhodolith/maërl bed thinning in S1; (**c**) furrows probably left by trawling in S2.

Boxwork was the most represented morphotype in the Lampedusa rhodolith/maërl bed, formed by the concretion of different overlapped algal species with numerous cavities filled by sediment; the dominance of boxwork morphotype is usually found in areas at low hydrodynamic regime [27,41]. Instead, branch and praline morphotypes accounted for a minor percentage, heterogeneously distributed within the sampling sites. In general, the distribution of rhodoliths is controlled by a combination of many environmental variables like temperature, salinity, irradiance, nutrients and water chemistry [42,43], as well as hydrodynamic regimes influencing the morphotype composition [35].

The studied rhodolith/maërl bed was also characterized by the presence of lamellar thalli mainly belonging to *Peyssonnelia* spp., indicating the occurrence of the facies of *Peyssonnelia rosa–marina* which is mainly present on fluid or mobile mud in turbid currents [44,45].

Overall, the examined bed was characterized by a rather diverse benthic assemblage, although it was relatively poor in macroalgal elements. Macroalgae, in fact, accounted for ~14% of the whole macrobenthic community in terms of number of taxa, while in the nearby Maltese bed was ~26% [7] and at Ustica Island (Southern Tyrrhenian Sea), 117 macroalgal taxa were found at comparable depths [46]. The epiflora found in Lampedusa Island showed the dominance of Rhodophyceae, confirming the ability of this group to adapt to great depths. Crustose algae and algae with coriaceous thalli, resistant to mechanical and biological disturbances like *Pneophyllum confervicola*, *Zonaria tournefortii*, *Cutleria chilosa*, *Hydrolithon farinosum*, *Osmundaria volubilis*, *Peyssonnelia dubyi*, were the most frequent.

The benthic assemblage was characterized by high biodiversity, except for the macroalgal component, and significant differences between sampling sites were found. Although pairwise comparisons were not significantly different, the species richness and number of

specimens in sampling site S1 were lower than in the other sampling sites (Table 1). The presence in S1 of a large sandy patch almost completely devoid of rhodoliths, could be also responsible for the low algal biodiversity and the different macrobenthic assemblage compared to the other sampling sites. As already demonstrated by many authors, the presence of rhodoliths increases species richness and diversity in the benthic community [32,47,48] because of the creation of new microhabitats within and around the rhodoliths for many species. Sites with such high biodiversity can support high macrobenthic secondary production, and consequently affect the diversity and productivity of marine ecosystems [49,50].

The rhodolith/maërl bed biodiversity was mainly represented by Polychaeta and Crustacea as already discovered in other Mediterranean beds [28,51], as well as in Southern Australia [52], unlike the Atlantic ones where Crustacea dominated [53]. Furthermore, the high biodiversity was coupled with a greater abundance and dominance of deposit feeders and predators; the high species diversity of deposit feeders and predators might lead to the hypothesis of niche diversification among deposit feeders and richness of prey organisms for predators. This confirms what was reported by many authors, namely that the rhodolith/maërl bed, being a key point of energy and matter fluxes, contributes to the maintenance of marine biodiversity [54–57].

The macrozoobenthic assemblage of the studied rhodolith/maërl bed was composed of species belonging to different biocoenoses, the most represented being coastal detritic (with the species *Ditrupa arietina*, *Ebalia edwardsii*, *Modiolula phaseolina*, *Petta pusilla*, *Urothoe elegans*, *Vermiliopsis infundibulum*) and Coarse Sand and Fine Gravel under Bottom Current (with the species *Branchiostoma lanceolatum*, *Venus casina*) [58]. In fact, in the Mediterranean Sea, rhodolith associations and maërl facies are commonly found in infralittoral and circalittoral zones as part of coastal detritic and Coarse Sand and Fine Gravel under Bottom Currents [8,10].

The study of the benthic assemblage has also allowed the identification of non-indigenous species, considered as putative pressure on the rhodolith/maërl bed. Remarkable is the presence of the amphipod *Linguimaera caesaris*, the most abundant species of the overall macrozoobenthic assemblage. This species is a Lessepsian immigrant and it has been previously reported for Egypt, Israel, Libya, Türkiye [1], Cyprus, and Tunisia [59–63], sometimes as *Maera hamigera* or *Hamimaera hamigera* due to taxonomic misunderstandings. The species has not yet been included in the inventory of non-indigenous species for the Italian seas under the Marine Strategy Framework Directive [64]. Given the high number of individuals found, as well as the wide distribution of this species along the rhodolith/maërl bed, it is possible to consider *L. caesaris* as established. To date, the impact of this species on the benthic community is not known; however, its important presence compared to that of native species should represent a stimulus to consider an adequate monitoring of the rhodolith/maërl bed benthic community, in order to deepen the knowledge on the ecology of the species and possibly adopt management and conservation measures.

The spreading path of this species in the Mediterranean, on the one hand along the Levantine waters northward to the Aegean and on the other along the Southern Mediterranean coast in a westerly direction, agrees with the main route of other Lessepsian species range expansion [62,65]. Since this species (like other benthic amphipods) does not have pelagic larval stages, it can be assumed that it has reached Lampedusa Island via passive dispersal, probably from near Tunisia. Another Lessepsian species, the polychaete *Notomastus aberans*, is also present but in very low abundance.

Other abundant species of the macrozoobenthic assemblage are the gravellicole sea urchins *Sphaerechinus granularis* and *Echinocyamus pusillus*, together with species with a wide range of habitat preferences like *Glycera alba*, *Apseudes talpa*, *Cheirocratus sundevalli*, *Lysidice ninetta*. Such an assemblage is comparable to that found in other Mediterranean rhodolith/maërl beds, which include species typical of different biocoenoses [28,66].

To date, no species uniquely associated with rhodolith/maërl beds have been identified, but rather species found in the beds have diverse habitat preferences or belong to gravel communities. For this reason, some authors have stated that rhodolith/maërl beds

are characterized by a core set of species with high abundance and well-adapted to the high hydrodynamism, and some rare and less abundant species, more related to small-scale spatial patchiness [67].

In the Mediterranean, due to the great depths of rhodolith/maërl beds, the associated benthic community has received relatively little attention, although in the last years the studies have increased [28,68,69]. The study of their biodiversity has proved important to validate the health status of the bed, as well as to construct a species inventory baseline for future monitoring activities.

Notwithstanding the great biological importance of rhodolith/maërl beds as reservoirs of biodiversity, they suffer a variety of anthropogenic disturbances in the Mediterranean that include impacts due to fishing and chemical pollution by organic matter and excess of nutrients. From an ecosystem perspective, actions for their management and conservation, such as bans on the use of towed gears on rhodolith/maërl beds and measures to limit the impacts on water quality above rhodolith/maërl beds, should be greatly recommended. A higher conservation status for rhodolith habitats and maërl-forming species should also be considered in European legislation, as well as the designation of "no-take" reserves. Whatever management strategy is adopted, a program to monitor the "health" of European rhodolith/maërl beds and further research on this delicate habitat are fundamental for the conservation of this important ecological resource.

**Supplementary Materials:** The following supporting information can be downloaded at: https://www.mdpi.com/article/10.3390/jmse10121808/s1, Table S1: List of species/taxa detected on Lampedusa rhodolith/maërl bed.

**Author Contributions:** Conceptualization, T.M., P.P., M.F. and L.C.; Methodology, T.M., P.P., A.P., S.C. and L.C.; Formal Analysis, T.M., P.P., A.P., S.C. and L.C.; Investigation, T.M., A.P., S.C., M.F., A.M.M., A.A. and L.C.; Data Curation, T.M., P.P., A.P., S.C., A.M.M. and L.C.; Writing—Original Draft Preparation, T.M., P.P. and L.C.; Writing—Review & Editing, T.M., P.P., S.C., M.F., A.M.M. and L.C.; Visualization, P.P.; Supervision, M.F. and L.C.; Project Administration, M.F. and L.C.; Funding Acquisition, L.C. All authors have read and agreed to the published version of the manuscript.

**Funding:** This research was funded by Interreg V–A Italia–Malta Project [C1–3.1–31], HARMONY "Italo–Maltese harmonization for a good state of the environment: integrity of the seabed and interaction with invasive species to preserve the functioning of marine ecosystems".

**Institutional Review Board Statement:** Not applicable.

**Informed Consent Statement:** Not applicable.

**Data Availability Statement:** The data presented in this study are available in the manuscript and Supplementary Material.

**Acknowledgments:** The authors thank the R/V ASTREA crew for their kind help during the survey activities.

**Conflicts of Interest:** The authors declare no conflict of interest.

## Notes

[1] The United Nations officially recognised Turkey renaming as Türkiye after the government formally advocated for the new name.

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
