# Peer review of "Sneaking into a Hotspot of Biodiversity: Coverage and Integrity of a Rhodolith Bed in the Strait of Sicily (Central Mediterranean Sea)"

_jmse, doi:10.3390/jmse10121808_

Round 1

Reviewer 1 Report

References specified in the Habitats Directive (92/43/CEE), the EC Council Regulation 1967/2006 and the Marine Strategy Framework Directive (MSFD) (2008/56/EC) in Lines 81-83 should be numbered and added to the References.

Lines 219 and 225; 2 in km2 should be written as superscript.

Line 435; It would be appropriate to write Türkiye instead of Turkey. This situation was stated in the official gazette and the United Nations accepted it.

Author Response

dear reviewer

thank you for your kind suggestions; we had the text revised by a english native speaker colleague as suggested. Please find below answer to your requests

References specified in the Habitats Directive (92/43/CEE), the EC Council Regulation 1967/2006 and the Marine Strategy Framework Directive (MSFD) (2008/56/EC) in Lines 81-83 should be numbered and added to the References.

  • We added the references requested

Lines 219 and 225; 2 in km2 should be written as superscript.

  • done

Line 435; It would be appropriate to write Türkiye instead of Turkey. This situation was stated in the official gazette and the United Nations accepted it.

  • we wrote the new name of Turkey and we added a note to clarify it

Reviewer 2 Report

This work on rhodolith bed is need of an hour. 

Author Response

dear reviewer

thank you for reading our manuscript. We had the text revised by a native english  speaker colleague as suggested

Reviewer 3 Report

Sneaking into a hotspot of biodiversity: coverage and integrity of a rhodolith bed in the Strait of Sicily (Central Mediterranean Sea) by Maggio et al describes the importance of the biodiversity in the strait of Sicily.

The manuscriprt is designed well and the results were presented in good shape and I believe the manuscript can be improved slightly by intrducing the biodiversity reachness (especially for crustaceans) of the region could be very valuable for example for loggerhead sea turtles. The other point is the monitoring of the invasive species are very important for the entire Mediterranean and such studies can be very valuable in this purposes. I think these two important remarks can be added whereever is appropriate.

Author Response

dear reviewer 

thank you for your kind suggestion; we added the remarks on biodiversity richness and NIS monitoring in the discussion section, as suggested. 

Reviewer 4 Report

Dear Authors

Thank you for conduct the research and for preparing this interesting manuscript using a diverse of methods such as ROVs, multibeams and grabs. I enjoyed reading your manuscript, and found it relevant to the audience and scope of this journal. I have suggested few editorial changes to the manuscript in the attached markup document to help improve the quality of the manuscript.

All the best with the publication.

Author Response

dear reviewer

thank you for your suggestions; please find our aswers in the attached file 
